# Nitrogen Removal Ability and Characteristics of the Laboratory-Scale Tidal Flow Constructed Wetlands for Treating Ammonium-Nitrogen Contaminated Groundwater

**Amit Kumar Maharjan [1], Kazuhiro Mori [2] and Tadashi Toyama [2],***

[1] Integrated Graduate School of Medicine, Engineering and Agricultural Sciences, University of Yamanashi, Yamanashi 400-8511, Japan; g17dea01@yamanashi.ac.jp

[2] Graduate Faculty of Interdisciplinary Research, University of Yamanashi, Yamanashi 400-8511, Japan; mori@yamanashi.ac.jp

* Correspondence: ttohyama@yamanashi.ac.jp

**Abstract:** Constructed wetlands (CWs) are an effective technology to remove organic compounds and nitrogen (N) from wastewaters and contaminated environmental waters. However, the feasibility of CWs for ammonium-N ($NH_4^+$-N)-contaminated groundwater treatment is unclear. In this study, zeolite-based laboratory-scale CW was operated as a tidal flow CW with a cycle consisting of 21-h flooded and 3-h rest, and used to treat $NH_4^+$-N (30 mg $L^{-1}$) contaminated groundwater. In addition to $NH_4^+$-N, nitrite ($NO_2^-$-N) and nitrate ($NO_3^-$-N) were also not detected in the effluents from the tidal flow CW. The N removal constant remained high for a longer period of time compared to the continuous flow CW. The higher and more sustainable N removal of the tidal flow CW was due to the in-situ biological regeneration of zeolite $NH_4^+$-N adsorption capacity. Vegetation of common reeds in tidal flow zeolite-based CW enhanced nitrification and heterotrophic denitrification activities, and increased the functional genes of nitrification (AOB-*amoA* and *nxrA*) and denitrification (*narG*, *nirK*, *nirS*, and *nosZ*) by 2-3 orders of magnitude, compared to CW without vegetation. The results suggest that the combination of zeolite substrate, tidal flow, and vegetation is key for the highly efficient and sustainable N removal from $NH_4^+$-N contaminated groundwater.

**Keywords:** nitrogen removal; $NH_4^+$-N contaminated groundwater; constructed wetland; tidal flow; zeolite; nitrification; denitrification; biological regeneration

## 1. Introduction

Groundwater is a major source of drinking water supply, with 50% of drinking water supplies in the world based on groundwater [1]. However, groundwater can often be contaminated with ammonium-nitrogen ($NH_4^+$-N). Heavy $NH_4^+$-N contamination of groundwater has been reported in certain parts of Australia (up to 120 mg $NH_4^+$-N $L^{-1}$) [2], Vietnam (up to 69.8 mg $NH_4^+$-N $L^{-1}$) [3], China (up to 10 mg $NH_4^+$-N $L^{-1}$) [4], Central India (up to 57 mg $NH_4^+$-N $L^{-1}$) [5], and the Kathmandu Valley of Nepal (up to 57.3 mg $NH_4^+$-N $L^{-1}$) [6]. Excess $NH_4^+$-N in groundwater makes it undrinkable due to bad taste and odor, reduces chlorine disinfection, and increases the possibility of pathogenic contamination during water distribution. Hence, reducing $NH_4^+$-N concentration of the groundwater prior to the conventional treatment and distribution of drinking water is essential.

The in-situ permeable reactive barrier (PRB) has been recognized as an effective technology for removing $NH_4^+$-N from groundwater [4,7,8]. However, it requires large-scale construction and incurs a high initial cost [9]. Hence, alternative $NH_4^+$-N removal techniques requiring minimum energy

and low operational/construction cost are needed, particularly in developing countries and small rural communities.

Constructed wetlands (CWs) are engineered wetland systems designed to stimulate the natural processes based on interactions among substrate media, microorganisms, and plants for the treatment of wastewaters. CWs have the advantages of low energy consumption and low costs, reduced and easy maintenance, are environmentally friendly, and have a high potential for application in developing countries and small rural communities [10–12]. Originally, CWs were designed and operated for the removal of organic matter, suspended solids, nitrogen, and phosphorus from wastewater [13–15]. CWs have also been applied for remediation of contaminated groundwater with nitrate-nitrogen ($NO_3^-$-N) [16], chlorinated solvents [17,18], benzene, and methyl-*tert*-butyl-ester (MTBE) [19,20]. However, there are very few studies investigating $NH_4^+$-N removal from contaminated groundwater by CWs. Seeger et al. [21] reported the performance of a CW for treating a groundwater multiple-contaminated with 20 mg $L^{-1}$ benzene (99% removal), 3.7 mg $L^{-1}$ MTBE (82% removal), and 45 mg $L^{-1}$ $NH_4^+$-N (54% removal). The potential of CWs to remove $NH_4^+$-N from contaminated groundwater, and the process for highly efficient and sustainable $NH_4^+$-N removal, are still unclear.

In natural freshwater sediments and CWs, microorganism-mediated nitrification and denitrification are the major pathways for $NH_4^+$-N removal [22–24]. Nitrification and denitrification require aerobic and anaerobic conditions, respectively, which makes it difficult to promote both nitrification and denitrification in a single CW [25]. To address this issue, tidal flow CWs operated with a repeated cycle consisting of fill, contact (flooded), drain, and rest period have recently attracted attention [26]. During the drain and rest period, air can be drawn into substrate media from the atmosphere, following which substrate media can have aerobic conditions. During the flooded period, air can be released from the media and consumed by microbial reactions, resulting in anaerobic conditions in the media and water phase. Tidal flow CWs thus can provide appropriate aerobic/anaerobic conditions for both nitrification and denitrification processes [27,28]. In addition to the flow type, substrate media can also affect $NH_4^+$-N removal in CWs. Zeolite, with high porosity and high cation exchange capacity (especially for $NH_4^+$-N), has been recognized as an ideal substrate material in CWs [29–31]. For sustainable and effective use, regeneration of zeolite adsorbed with $NH_4^+$-N is necessary. Tidal flow CWs enable the in-situ biological regeneration of zeolites due to the $NH_4^+$-N removal via stimulated nitrification/denitrification. If this hypothesis is verified, the tidal flow and zeolite-based CWs will offer a highly efficient and sustainable way to remove $NH_4^+$-N from the contaminated groundwater. However, to the best of our knowledge, there are no studies clearly showing the potential to remove $NH_4^+$-N from contaminated groundwater by tidal flow CWs.

This study investigates the removal efficiency of $NH_4^+$-N from contaminated groundwater by a tidal flow, zeolite-based CW. Laboratory-scale tidal flow zeolite-based CWs were set up and used to treat synthetic groundwater contaminated with 30 mg $L^{-1}$ of $NH_4^+$-N. The advantages of nitrogen (N) removal, including $NH_4^+$-N adsorption on zeolite, nitrification and denitrification by the tidal flow CW over continuous flow CW, and the effect of vegetation on these N removal functions in CWs, are discussed below.

## 2. Materials and Methods

### 2.1. Synthetic $NH_4^+$-N Contaminated Groundwater

A synthetic groundwater containing 30 mg $L^{-1}$ of $NH_4^+$-N was prepared and used in this study. The composition of synthetic groundwater included: $Na_2HPO_4 \cdot 12H_2O$ (104.5 mg $L^{-1}$), $KH_2PO_4$ (17 mg $L^{-1}$), NaCl (37.5 mg $L^{-1}$), KCl (17.5 mg $L^{-1}$), $CaCl_2 \cdot 2H_2O$ (23 mg $L^{-1}$), $MgSO_4 \cdot 7H_2O$ (25.6 mg $L^{-1}$), $NaHCO_3$ (353 mg $L^{-1}$), and $(NH_4)_2SO_4$ (141.6 mg $L^{-1}$). The constituents and their concentrations in the synthetic groundwater were determined on the basis of the chemical composition of $NH_4^+$-N-contaminated groundwater sampled from the Kathmandu Valley, Nepal [32]. In addition, the $NO_3^-$-N contaminated groundwater with 30 mg $L^{-1}$ of $NO_3^-$-N (182.12 mg of $NaNO_3$ per L,

instead of $(NH_4)_2SO_4$ in the above $NH_4^+$-N contaminated groundwater) was prepared and used in the denitrification experiment.

## 2.2. Experimental Setup and Conditions

### 2.2.1. N Removal from $NH_4^+$-N Contaminated Groundwater by Tidal Flow CWs and Continuous Flow CWs

Here, we investigated the effect of the flow type, i.e., tidal and continuous flows, on the N removal abilities of CWs. Two types of laboratory-scale CWs: (i) tidal flow, and (ii) continuous flow, were set up in duplicate inside a greenhouse without artificial lights and temperature controllers at the University of Yamanashi, Kofu, Yamanashi, Japan. For each CW, a plastic column (150 mm diameter × 650 mm height) with an outflow port at the bottom was used. The columns were filled with pumice rock (grain size of about 10 mm) from bottom to 20 mm height, and zeolite (grain size of 3–5 mm), from 20 to 600 mm height. Common reed seedlings (*Phragmites australis*; 20 numbers; 700–900 mm tall) were planted in each CW. The side surface parts of the CWs were wrapped with silver color plastic sheets to block sunlight. The configurations of the two types of CW are shown in Figure 1. The CWs were set up at the end of July, and subjected to a 2-week start-up period for the inoculation and colonization of microorganisms into CW. During the start-up period, each CW was operated in a tidal flow (21-h flooded and 3-h rest) for 2 weeks. In the first week, each CW was filled with 4 L of synthetic groundwater mixed with activated sludge (100:1, v/v). In the following week, each CW was filled with 4 L of synthetic groundwater. The activated sludge was collected from the activated sludge settling tank at a conventional municipal wastewater treatment plant in Kofu, Yamanashi, Japan. Following the start-up period, two CWs were operated as tidal flow CWs (21-h flooded with 4 L of synthetic groundwater, and 3-h rest). Tidal flow (inflow and outflow) was controlled by a pump (Masterflex L/S; Cole-Parmer, IL, US) with a timer. The other two CWs were operated as continuous flow CWs (4 L d$^{-1}$) with a pump, as a control experiment. The N removal experiments using the tidal flow and continuous flow CWs continued for 105 days from mid-August to late November. Influent and effluent samples were collected from all CWs, and their $NH_4^+$-N, $NO_2^-$-N, and $NO_3^-$-N concentrations were determined.

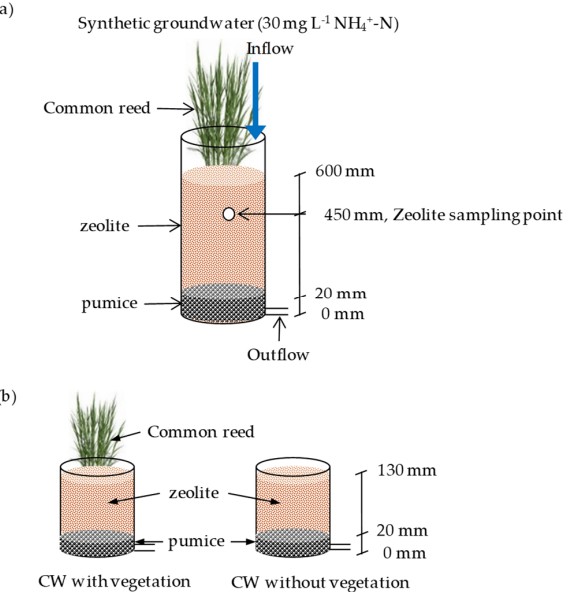

**Figure 1.** (**a**) Laboratory-scale zeolite-based tidal flow and continuous flow constructed wetlands (CWs) with vegetation (common reed plants), to study the effect of flow type in N removal and (**b**) smaller-scale tidal flow CWs, with and without vegetation, to study the effect of vegetation.

A flowchart of this experiment is shown in Figure S1. During the experimental period, N removal potential of zeolite materials in both CWs was monitored every two weeks, as follows. Zeolite materials (100 g) were collected from each CW at 150 mm depth sampling point (Figure 1) and transferred into a 200 mL flask. The flasks were filled with 100 mL of synthetic $NH_4^+$-N contaminated groundwater (30 mg $L^{-1}$) and incubated at 120 rpm and 25 °C for 24 h. The groundwater samples were collected every few hours from flasks; and their $NH_4^+$-N, $NO_2^-$-N, and $NO_3^-$-N concentrations were determined. The results were used for the calculation of the kinetic constant of $NH_4^+$-N adsorption.

### 2.2.2. Evaluation of Vegetation Effects on N Removal Ability in the Tidal Flow CW

We examined the effects of vegetation on N removal ability and characteristics in the tidal flow CW. A flowchart of this experiment is shown in Figure S1. The smaller-scale tidal flow CWs with and without vegetation were prepared with four 1 L-capacity plastic beakers (100 mm diameter, 140 mm height; Figure 1) with a drain port at the bottom. The beakers were filled with pumice rock (grain size of about 10 mm) from bottom to 20 mm height, and zeolite (grain size of 3–5 mm), from 20 to 130 mm height. Common reed seedlings (10 numbers; 500–600 mm tall) were planted in each of the two CWs (CWs with vegetation). The other two CWs were not vegetated with common reed seedlings (CWs without vegetation). To collect water samples, a 10-mm diameter sampling port was made at 100 mm depth from the surface level. The side surface parts of the CWs were wrapped with silver color plastic sheets to block sunlight. At first, a 2-week start-up period was set to inoculate and colonize microorganisms into substrate media in CWs. During the start-up period, each CW was operated in the tidal flow (21-h flooded and 3-h rest). In the first week, each CW was filled with 0.5 L of synthetic groundwater mixed with activated sludge (100:1, v/v). In the following week, each CW was filled with 0.5 L of synthetic groundwater. Following the start-up period, four CWs were operated in a tidal flow (21-h flooded with 0.5 L of synthetic groundwater, and 3-h rest) for 60 days, from the first of September to the end of October. The tidal flow (inflow and outflow) was controlled by a pump (Masterflex) with a timer. On the 60th day, water samples were collected every few hours after filling from the sampling port of all CWs, and their $NH_4^+$-N, $NO_2^-$-N, and $NO_3^-$-N concentrations were determined. Furthermore, zeolite materials were collected from the CWs, both with and without vegetation, in order to analyze bacterial community.

### 2.2.3. Denitrification Experiments Using Common Reed Roots from Tidal Flow CWs

The effect of common reed roots on denitrification in CWs was examined. A flowchart of this experiment is shown in Figure S1. Zeolite (100 g) was collected from the tidal flow CWs with and without vegetation on the 60th day (Section 2.2.2) and transferred into a 100 mL vial, and 5 g (wet) of common reed roots was collected from the CWs with vegetation and added to the vial. Synthetic $NO_3^-$-N contaminated groundwater (50 mL), containing 30 mg of $NO_3^-$-N $L^{-1}$, was added to each vial. The vials with and without common reed roots were prepared in triplicate. All vials were purged with nitrogen ($N_2$) gas for 2 min to create anaerobic conditions, and the vials were closed with a butyl-rubber and aluminum cap. All vials were incubated at 120 rpm and 25 °C for 24 h. Water samples were collected from the vials every few hours, and their $NH_4^+$-N, $NO_2^-$-N, and $NO_3^-$-N concentrations were determined.

### 2.3. Analysis of Samples

Water samples were filtered through a membrane filter (polypropylene, pore size = 0.45 μm; Membrane Solutions Co. Ltd., Minato-ku, Japan). N concentrations in the water samples were measured in accordance with the Standard Methods for the Examination of Water and Wastewater [33]. Concentrations of $NH_4^+$-N, $NO_2^-$-N, and $NO_3^-$-N were determined by indophenol method, *N*-(1-naphthyl) ethylenediamine method, and ultraviolet spectrophotometric screening method, respectively, with a spectrophotometer (UVmini-1280; Shimadzu Co. Ltd., Japan).

N removal efficiency in the tidal flow and continuous flow zeolite-based CWs (Section 2.2.1) was calculated using Equation (1).

$$N\ removal\ efficiency = \left(1 - \frac{Effluent - N}{Influent - N}\right) \times 100\%$$ (1)

where *Influent-N* and *Effluent-N* represent the sum of $NH_4^+$-N, $NO_2^-$-N, and $NO_3^-$-N in influent and effluent, respectively.

First-order $NH_4^+$-N removal rate in N removal potential experiment (Section 2.2.1) was calculated as Equation (2).

$$First - order\ removal\ rate = \left(\frac{Ln\ C_1 - Ln\ C_2}{t_1 - t_2}\right)$$ (2)

where $C_1$ and $C_2$ are the concentrations of $NH_4^+$-N at time $t_1$ and $t_2$, respectively.

## 2.4. Microbial Community Analyses

A flowchart of microbial community analyses is shown in Figure S1. To detach the microorganisms from the zeolite substrates, zeolite (2 g) was weighed and placed in a 15 mL tube. Following this, each sample was vortexed for 1 min with 4 mL of phosphate buffer saline (1.44 g $L^{-1}$ $NaH_2PO_4$, 0.24 g $L^{-1}$ $K_2HPO_4$, 8 g $L^{-1}$ NaCl, 0.2 g $L^{-1}$ KCl; pH 7.4), and shaken for 1 min. The suspension was passed through a membrane filter (pore size 0.2 µm; mixed cellulose esters membrane; Merck Millipore). The total DNA of the microorganisms on the membrane filter was extracted by using the Nucleo-spin soil kit (MACHEREY-NAGEL GmbH, Duren, Germany), according to the manufacturer's protocol. Bacterial 16S rRNA and the ammonia monooxygenase (*amoA*) of ammonia-oxidizing bacteria (AOB), nitrite oxidoreductase (*nxrA*), nitrate reductase (*narG*), nitrite reductase (*nirK* and *nirS*), and nitrous oxide reductase (*nosZ*) genes were quantified by real-time quantitative polymerase chain reaction (RT-qPCR) in a Thermal Cycler Dice RealTime System II (Takara Bio Inc., Shiga, Japan). Each 25 µL reaction mixture contained 12.5 µL of SYBR Premix Ex Taq (TaKaRa Bio), 0.5 µM of each forward and reverse primer (Table S1), 2 µL of template DNA, and 9.5 µL of deionized $H_2O$. The qPCR reaction conditions were as follows: initial denaturation by pre-heating at 95 °C for 30 s, 40 cycles at 98 °C for 5 s, annealing at the specified temperatures (which varied with primer type [34–40]; Table S1) for 50 s, and an extension at 72 °C for 1 min, followed by a dissociation stage (95 °C for 15 s, 60 °C for 30 s, and 95 °C for 15 s). A standard curve was plotted for each gene using a synthetic plasmid carrying the target sequence. All qPCRs were conducted in triplicate, and the average gene abundances in the substrates (copies $g^{-1}$ of the zeolite substrate) were calculated for each CW.

The extracted bacterial DNA samples were also subjected to Illumina MiSeq 16S rRNA gene sequencing (Illumina, San Diego, CA, USA). The V4 region of the 16S rRNA gene was amplified by PCR with the universal primers 515F (5'-Seq A-TGT-GCC-AGC-MGC-CGC-GGT-AA-3') and 806R (5'-Seq B-GGA-CTA-CHV-GGG-TWT-CTA-AT-3'). PCR amplicons were sequenced in an Illumina MiSeq Sequencer (Illumina, San Diego, CA, USA). Sequence reads were analyzed with Sickle v. 1.33 [41], Fastx Toolkit v. 0.0.13.2 [42], FLASH v. 1.2.10 [43], and USEARCH v. 8.0.1623_i86linux64 [44]. In these analyses, contigs were formed, and error sequences and chimeras were removed. All operational taxonomic units (OTUs) were clustered at a cutoff of 0.03 (97% similarity). Only phylogenetic groups accounting for more than 1% relative abundance in at least one of the datasets were listed (≤1% were summed as "others"). Sequencing and sequence-read analyses were conducted at FASMAC (Kanagawa, Japan).

## 2.5. Statistical Analysis

The mean and standard deviation (SD) of the physicochemical parameters and nitrogen concentrations were calculated. Gene abundances (±SD) in the zeolite substrate of each CW were

also calculated. A *t*-test was used to compare the pairs of groups for significant differences ($p < 0.05$). The data were processed in SPSS v. 20 (IBM Corp., Armonk, NY, USA).

## 3. Results

### 3.1. N Removal from $NH_4^+$-N Contaminated Groundwater by Tidal Flow and Continuous Flow CWs

The changes in N concentrations in influent and effluent, and the N removal efficiency of the tidal flow and continuous flow CWs over 105 days are shown in Figure 2. In the tidal flow CWs, $NH_4^+$-N and $NO_2^-$-N were not detected in the effluent throughout the experiment period. The $NO_3^-$-N concentration in the effluent ranged between 0.6 to 4.8 mg L$^{-1}$. The N removal efficiency of the tidal flow CW was 83.9%–98.2%. In the continuous flow CW, $NH_4^+$-N was not detected in the effluent for the first 21 days, and its concentration increased during the 21-63-day period, ranging between 6.4 to 7.9 mg L$^{-1}$. The $NO_2^-$-N was not detected in the effluent throughout the experiment period. The $NO_3^-$-N concentration in the effluent was between 0.4 to 3.5 mg L$^{-1}$ over 105 days. The N removal efficiency of the continuous flow CW was 67.6%–97.2%. After the 21st day, $NH_4^+$-N concentration in the effluent was significantly lower ($p < 0.05$) in the tidal flow CW, compared to the continuous flow CW, and the N removal by the tidal flow CW was significantly higher ($p < 0.05$) than that by the continuous flow CW.

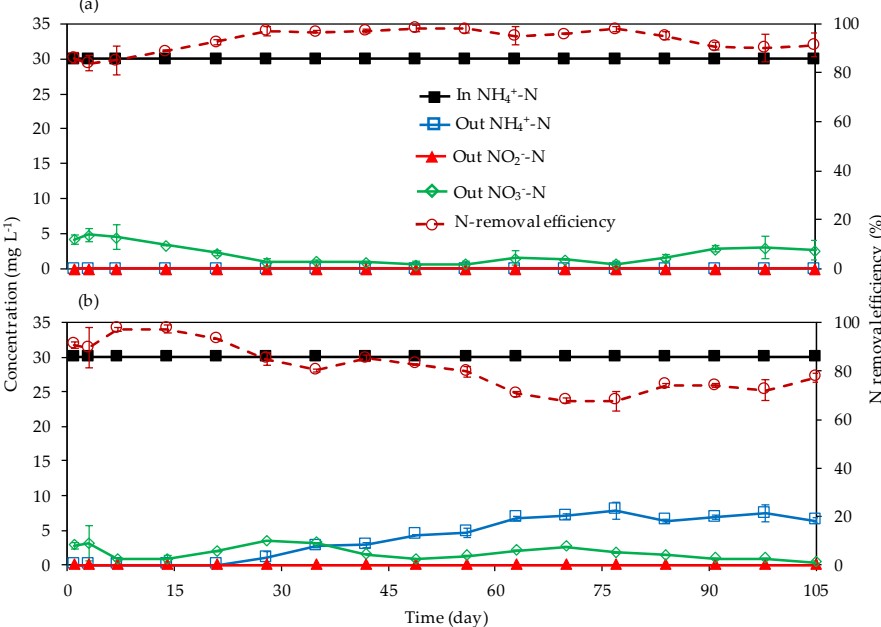

**Figure 2.** Changes in $NH_4^+$-N, $NO_2^-$-N, and $NO_3^-$-N concentration and N removal efficiency over 105 days of CW operation in (**a**) tidal flow, and (**b**) continuous flow. Values are means ± SD (n = 2).

The results of the N removal potential of zeolite-microbe association of both CWs are shown in Figure 3. The $NH_4^+$-N removal was due to adsorption on zeolite and nitrification by microorganisms. The results clearly show a higher N removal by zeolite-microbe association in the tidal flow CW, compared to that in the continuous flow CW. The changes in $NH_4^+$-N concentration during the first 2–3 h (Figure 3) were fitted to the first-order kinetic model (Figure S2). Because $NO_2^-$-N and $NO_3^-$-N were not generated during the first 2–3 h, $NH_4^+$-N must have been removed by adsorption on zeolite rather than nitrification ($NH_4^+$-N oxidation to $NO_2^-$-N and $NO_3^-$-N). Thus, the first-order $NH_4^+$-N removal rate can be considered as the $NH_4^+$-N adsorption rate on zeolite. The first-order $NH_4^+$-N adsorption rates are summarized in Figure 4. The $NH_4^+$-N adsorption kinetic constants of zeolite-microbe in both the tidal flow and continuous flow CWs decreased gradually. However, the $NH_4^+$-N adsorption rates of the tidal flow CWs were higher than those of the continuous flow CWs.

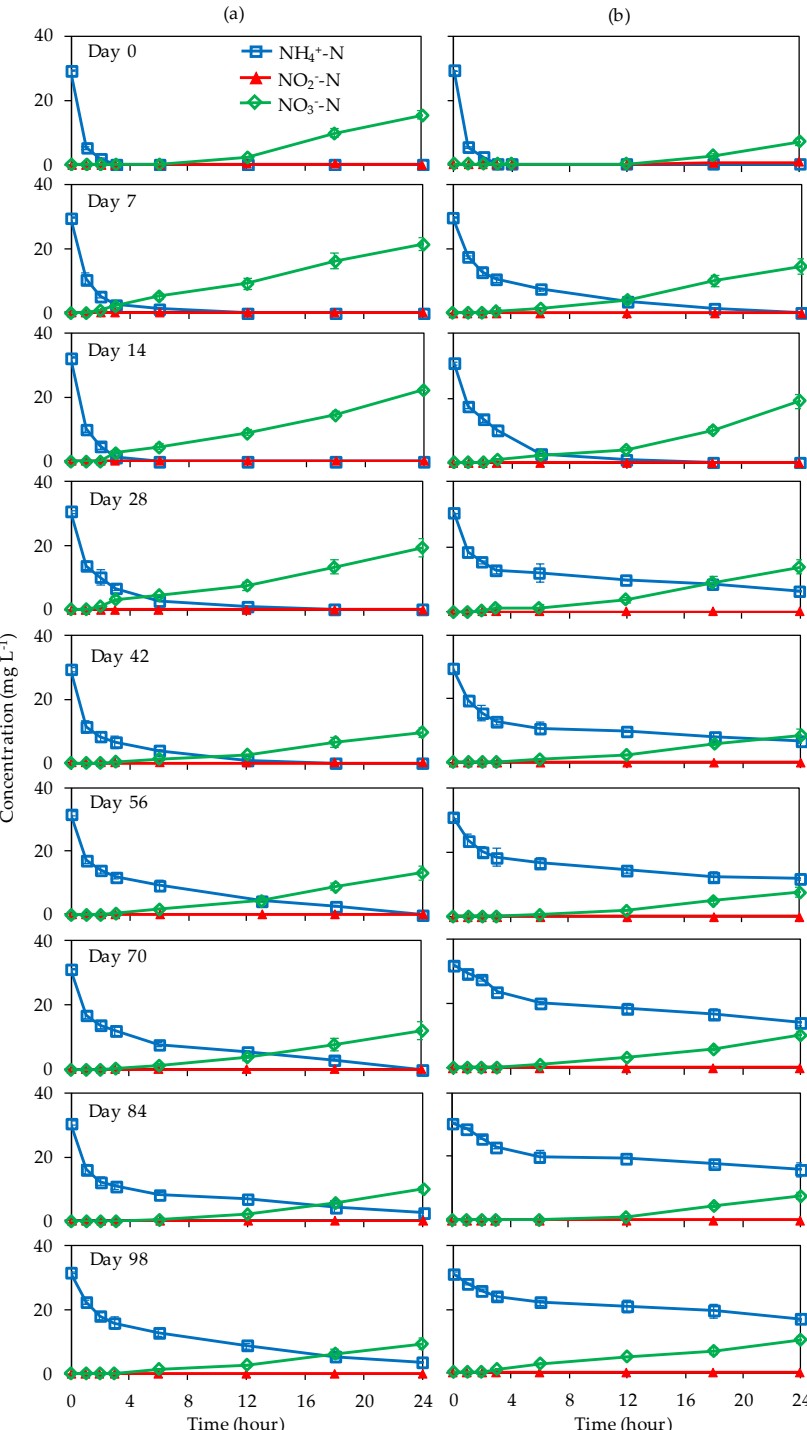

**Figure 3.** N removal potential of zeolite-microbe association in (**a**) tidal flow CWs, and (**b**) continuous flow CWs. Values are means ± SD (n = 3).

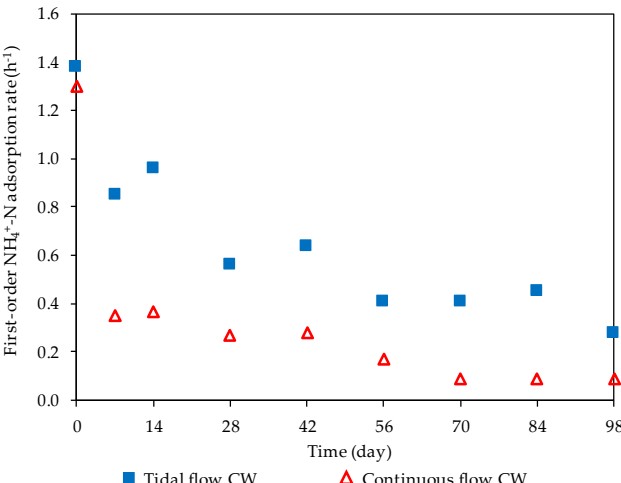

**Figure 4.** First-order $NH_4^+$-N adsorption rate on zeolites in tidal flow CWs and continuous flow CWs.

### 3.2. Evaluation of Vegetation Effects on N Removal Ability in the Tidal Flow CWs

The change of N concentrations over time in the tidal flow CWs with and without vegetation after 60 days of operation are shown in Figure 5. In the tidal flow CW with vegetation, $NH_4^+$-N concentration rapidly and completely decreased within 3 h. The $NO_3^-$-N concentration rapidly increased for 2 h, and then gradually and completely decreased within 18 h. The $NO_2^-$-N concentration was not detected for 24 h. In the tidal flow CW without vegetation, $NH_4^+$-N concentration decreased rapidly and completely within 9 h. The $NO_2^-$-N and $NO_3^-$-N concentrations increased for 3 h, following which $NO_2^-$-N concentration decreased; however, the $NO_3^-$-N concentration increased gradually for 24 h. The difference in the dynamics of $NO_3^-$-N concentration between CWs with and without vegetation might be due to the occurrence of denitrification in CWs. Previous studies have shown that the contribution of plant uptake to N removal in CWs is lesser than biological nitrification/denitrification [23,45].

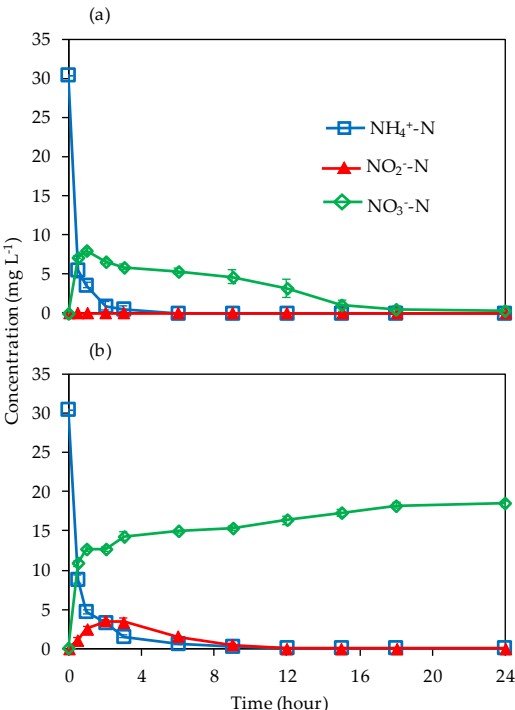

**Figure 5.** Changes in N concentrations in groundwater inside tidal flow CWs: (**a**) with vegetation (common reed plants), and (**b**) without vegetation. Values are means ± SD (n = 3).

### 3.3. Denitrification Experiments Using Common Reed Roots from Tidal Flow CWs

The changes in N concentrations in the effluent of denitrification experiments in zeolite-microbe with common reed roots and zeolite-microbe without roots are shown in Figure 6. In zeolite-microbe with roots, $NO_3^-$-N concentration decreased completely within 24 h, whereas it decreased slightly, from 30 to 24 mg $L^{-1}$, in zeolite-microbe without roots. The results suggest that common reed roots supported denitrification.

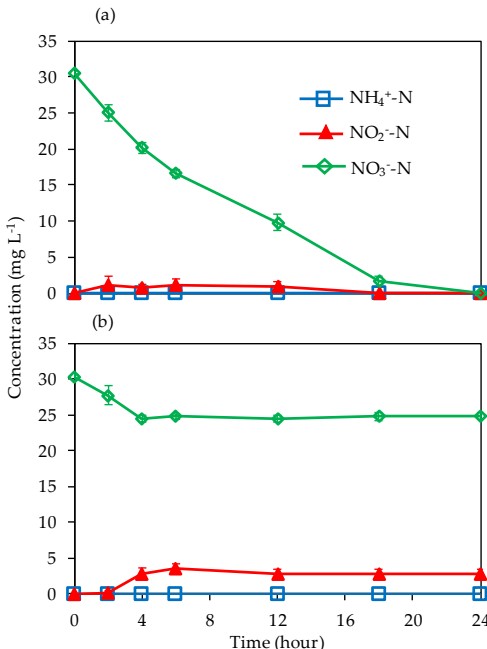

**Figure 6.** Changes in N concentrations over 24 h in denitrification experimental vial containing zeolite: (**a**) with common reed roots, and (**b**) without common reed roots, from the tidal flow CWs with and without vegetation, respectively, on the 60th day of operation. Values are means ± SD (n = 3).

### 3.4. Characterization of Microbial Communities in Tidal Flow CWs with and without Vegetation

The abundances of bacterial 16S rRNA and AOB-*amoA*, *nxrA*, *narG*, *nirK*, *nirS*, and *nosZ* genes on the 60th day of tidal flow CWs with and without vegetation are shown in Table 1. The abundance of 16S rRNA gene, nitrifying (AOB-*amoA* and *nxrA*), and denitrifying (*narG*, *nirK*, *nirS*, and *nosZ*) functional genes were about two-three orders of magnitude higher in the tidal flow CW with vegetation than that without vegetation.

**Table 1.** Abundance of bacterial 16S rRNA and ammonia oxidizing bacteria (AOB)-*amoA*, *nxrA*, *narG*, *nirK*, *nirS*, and *nosZ* genes in zeolite-based tidal flow CWs with and without vegetation (common reed plants) on the last day (i.e., 60th day) of operation. Average numbers of gene copies ±SD are shown for triplicate experiments.

| Target Gene | Abundance (copies $g^{-1}$ of zeolite) | |
| :---: | :---: | :---: |
| | with Vegetation | without Vegetation |
| Bacterial 16S rRNA | $(6.6 \pm 0.8) \times 10^{10}$ | $(4.0 \pm 0.4) \times 10^8$ |
| AOB-*amoA* | $(2.6 \pm 0.9) \times 10^8$ | $(2.3 \pm 0.3) \times 10^6$ |
| *nxrA* | $(7.1 \pm 2.6) \times 10^4$ | $(7.4 \pm 1.2) \times 10^2$ |
| *narG* | $(8.6 \pm 3.6) \times 10^8$ | $(5.5 \pm 0.5) \times 10^6$ |
| *nirK* | $(1.4 \pm 0.6) \times 10^8$ | $(1.1 \pm 0.4) \times 10^6$ |
| *nirS* | $(3.1 \pm 1.1) \times 10^8$ | $(1.0 \pm 0.6) \times 10^6$ |
| *nosZ* | $(1.9 \pm 0.1) \times 10^8$ | $(5.8 \pm 0.4) \times 10^4$ |

The bacterial community structures of the tidal flow CWs with and without vegetation at phylum, class, and order levels are shown in Figure 7. At phylum level, *Proteobacteria* (45.7% of all phyla), *Bacteroidetes* (14.4%), and *Planctomycetes* (6.5%) were dominant in the tidal flow CW with vegetation; *Proteobacteria* (49.2%), *Bacteroidetes* (25.8%), and *Verrucomicrobia* (5.8%) were dominant in the tidal flow CW without vegetation. At class level, *Alphaproteobacteria* (19.6% of all classes), *Betaproteobacteria* (10.8%), *Gammaproteobacteria* (10.1%), and *Saprospirae* (6.9%) were dominant in the tidal flow CW with vegetation; *Betaproteobacteria* (27.0%), *Saprospirae* (21.0%), *Alphaproteobacteria* (14.4%), and *Gammaproteobacteria* (5.8%) in the tidal flow CW without vegetation.

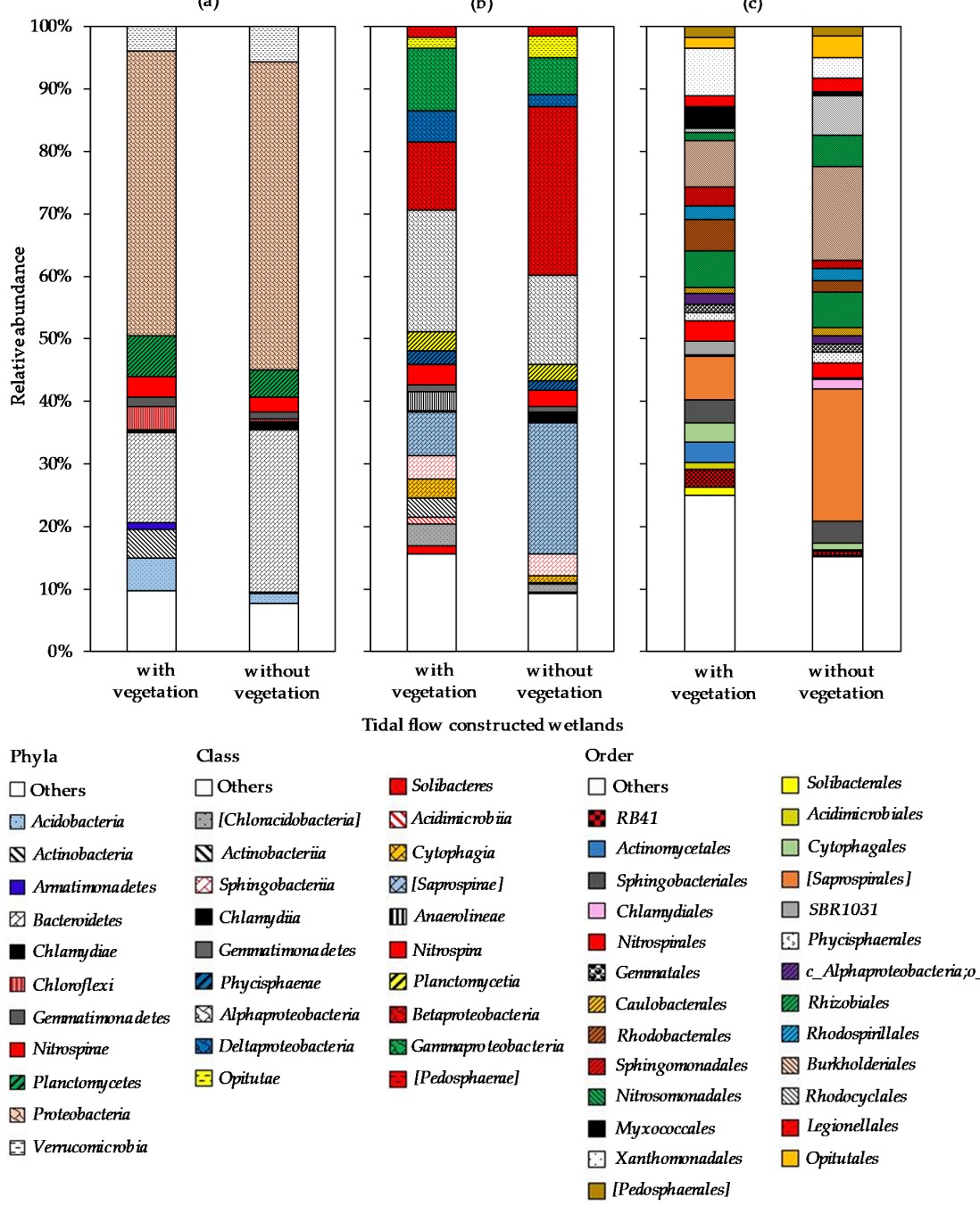

**Figure 7.** Bacterial community compositions at the (**a**) phylum, (**b**) class, and (**c**) order level in the tidal flow CWs with and without vegetation (common reed plants) on the last day (60th day) of operation.

Based on the bacterial order, the dominant groups were *Xanthomonadales* (7.7% of all orders), *Burkholderiales* (7.5%), *Saprospirales* (6.9%), and *Rhizobiales* (5.9%) in the CW with vegetation, and *Saprospirales* (21.0%), *Burkholderiales* (15.1%), *Rhodocyclales* (6.3%), and *Rhizobiales* (5.6%) in the CW without vegetation. The relative abundances of *Nitrosomonadales* (AOB) were 1.2% and 4.9% in the CWs with and without vegetation, respectively. Further, the relative abundances of *Nitrospirales* (nitrite-oxidizing bacteria; NOB) were 3.3% and 2.5% in the CWs with and without vegetation, respectively. The bacterial community structure, and the relative abundances of *Nitrosomonadales* and *Nitrospirales* in the tidal flow CWs with and without vegetation were similar.

## 4. Discussion

In this study, the ability and characteristics of zeolite-based tidal flow CWs to remove N from $NH_4^+$-N contaminated groundwater were examined. The tidal flow CW was operated at a cycle consisting of 21-h flooded and 3-h rest, and removed $NH_4^+$-N completely and repeatedly from $NH_4^+$-N contaminated groundwater (30 mg $L^{-1}$) over 105 days (Figure 2). In addition to $NH_4^+$-N, $NO_2^-$-N and $NO_3^-$-N were also not detected in the effluents from the tidal flow CW (Figure 2). The highly efficient N removal from $NH_4^+$-N contaminated groundwater and higher first-order kinetic constant for the $NH_4^+$-N adsorption remained high for a longer period of time, compared to the continuous flow CWs (Figures 2 and 4). Although zeolite has a high ability to remove $NH_4^+$-N, there is a limitation to its adsorption capacity [23]. For sustainable $NH_4^+$-N removal by zeolite, several methods for zeolite regeneration by brine solution treatment [46], NaOH treatment [47], electrochemical treatment [48], and biological nitrification treatment [49,50] have been reported. Tidal flow CWs have a higher atmospheric air supply and enhanced nitrification and denitrification, compared to continuous flow CWs [51,52]. Oxygen can be rapidly replenished into CW beds by the rhythmic tidal flow, and then dissolved oxygen (DO) is consumed by microbial activities for organic carbon degradation and nitrification along the depth [53,54] and contacting time [55,56]. After that, CW beds can change into anaerobic conditions. Unfortunately, DO concentration and oxidation-reduction potential were not monitored in this study. Enhanced nitrification and denitrification might have resulted in the in-situ biological regeneration of zeolite in the tidal flow CWs. Therefore, the tidal flow zeolite-based CW could remove N sustainably from $NH_4^+$-N contaminated groundwater.

In the zeolite-based tidal flow CWs, vegetation with common reeds accelerated complete nitrification without accumulation of $NO_2^-$-N in the CW with vegetation, compared to the CW without vegetation (Figure 5). Aquatic plants can release oxygen from roots and create aerobic conditions around roots and rhizosphere, increasing the activity and population of nitrifying bacteria [57]. In this study, the abundances of nitrifying functional genes (AOB-*amoA* and *nxrA* genes) were also two orders of magnitude higher in the CW with vegetation than those in the CW without vegetation (Table 1).

In addition, vegetation with common reeds significantly stimulated denitrification in the tidal flow CW. The decrease in $NO_3^-$-N concentration was significantly higher in the CW with vegetation and in denitrification experimental vial with roots than in the CW without vegetation and denitrification vial without roots, respectively (Figures 5 and 6). Aquatic plants, including common reeds, can release not only oxygen, but also organic compounds from roots into the rhizosphere [45,58]. The released organic compounds from plants can act as electron donors for heterotrophic denitrification bacteria, stimulating their activity and growth. Previous studies revealed the enhanced denitrification and higher abundance of denitrification bacteria in the CW [59] and sediment [60] with vegetation. In this study, denitrification ability and denitrifying functional genes (*narG*, *nirK*, *nirS*, and *nosZ* genes) were increased by vegetation with common reeds. The denitrification-stimulating effects of vegetation will be important and effective in N removal from contaminated groundwater lacking organic compounds. Although the vegetation in the CW increased the abundances of total bacteria, nitrification bacteria, and denitrification bacteria by two-three orders of magnitude compared to the CW without vegetation, the vegetation could not dramatically change the bacterial community compositions in the CWs. *Proteobacteria* and *Bacteroidetes* are generally the most dominant phyla in CWs [61–63]. In this study, the two phyla were dominant

in both tidal flow CWs, with and without vegetation (Figure 7). *Nitrosomonadales*-like AOB and *Nitrospirales*-like NOB were also detected in both CWs, with and without vegetation, and might be one of key nitrification bacteria in the CWs.

Denitrification ability is widely spread in diverse phylogenetic groups. Various order of bacteria, like *Rhizobiales* (*Rhizobiaceae* group) and *Rhodocyclales* (genus; *Thauera* and *Dechloromonas*) have been found to show denitrification [63,64]. These were present in the CWs in this study. In addition, *Planctomycetes* were also present in the investigated CWs. However, their contribution to N removal as anaerobic ammonium oxidation (anammox) remained unclear.

CWs have been applied for wastewater treatment [13–15] and remediation of contaminated groundwater with $NO_3^-$-N, chlorinated solvents, benzene, and MTBE [16,17,20]. This study is the first to demonstrate that CWs have potential to remove N from $NH_4^+$-N contaminated groundwater. In particular, our results found that the combination of zeolite substrate, tidal flow, and vegetation in CW is important for highly efficient and sustainable N removal. Pilot-scale tidal flow CWs have been operated as pump- or siphon-driven for treatment of various wastewaters [26,62,65,66]. Like these studies, it will be necessary to evaluate the performance of tidal flow CW treating $NH_4^+$-N contaminated groundwater in a pilot- or full-scale system.

## 5. Conclusions

This study clearly demonstrated that zeolite-based tidal flow (21-h flooded and 3-h rest) CWs are highly efficient in removing total N from $NH_4^+$-N (30 mg $L^{-1}$) contaminated groundwater in 105 days. In contrast, continuous flow CWs did not retain the higher rate of N removal from the $NH_4^+$-N contaminated groundwater. The highly efficient and sustainable N removal in tidal flow CWs might be due to the regeneration of zeolite $NH_4^+$-N adsorption capacity. The presence of vegetation (common reed) in the CWs enhanced nitrification and heterotrophic denitrification and increased the populations of nitrifying and denitrifying bacteria. These results strongly indicate that the major mechanism of the efficient and stable N removal in the zeolite-based tidal flow CW was a two-step process: $NH_4^+$-N was initially adsorbed onto zeolite and, subsequently, the adsorbed $NH_4^+$-N was converted to $NO_2^-$-N and $NO_3^-$-N, and finally transferred to the atmosphere as $N_2$ by enhanced nitrification and denitrification in CWs. The combination of zeolite substrate, tidal flow, and vegetation in CW should thus provide highly efficient and sustainable N removal. The knowledge obtained from this study will be helpful for the practical application of CWs to the treatment of $NH_4^+$-N contaminated groundwater.

**Supplementary Materials:** The following are available online at http://www.mdpi.com/2073-4441/12/5/1326/s1, Figure S1. The experimental flowcharts of Section 2.2.1 (a), Section 2.2.2 (b), Section 2.2.3 (c), and Section 2.4 (d); Table S1. Target genes for qPCR analysis, primers and sequences, amplification sizes, and annealing temperatures; Figure S2. First-order kinetic models for the decrease in $NH_4^+$-N concentrations during the first 2-3 hours of N removal potential experiment in zeolite-microbe association of (a) tidal flow CWs, and (b) continuous flow CWs.

**Author Contributions:** All authors contributed to the study design. A.K.M. and T.T. performed the experiments. A.K.M. interpreted the results and prepared a draft of the manuscript. T.T. supervised the experiments, checked and interpreted the results, and corrected the draft of the manuscript. K.M. discussed the results and critically reviewed the manuscript. All authors have read and agreed to the published version of the manuscript.

**Funding:** This research received no external funding.

**Acknowledgments:** The authors would like to thank Otsuka, Kaneko, Yamada, and Koga for their support in the experiments and lab analysis. The authors are also grateful to the editor and anonymous peer reviewers for their valuable input.

**Conflicts of Interest:** The authors declare no conflict of interest.

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
