# Peer review of "Nitrogen Removal Ability and Characteristics of the Laboratory-Scale Tidal Flow Constructed Wetlands for Treating Ammonium-Nitrogen Contaminated Groundwater"

_water, doi:10.3390/w12051326_

Round 1
Reviewer 1 Report
In this study, the potential and the effectiveness of the CWs-based mechanism in removing NH4+-N from groundwater are well discussed and presented. I think that the main advantages of this technology include its sustainability and its low impact on the environment. This work deserves publication, but requires some revisions:
- The structure of the Materials and Methods section needs to be improved. I found it difficult to follow the preparation of the experimental material as well as clearly understand the analysis procedure. In order to enhance the understanding of the experimental setup, Figure S1 may be inserted within the manuscript (in Sections 2.2.1 and 2.2.2) and not as supplementary material. Moreover, I think that adding flowcharts in Sections 2.2.1, 2.2.2 and 2.4 could facilitate the reader to fully understand and follow the description of the various steps of the experiment.
- Lines 149 and 152: “One hundred mg” and “Fifty mL” should be replaced with numbers.
- Results derived from the laboratory-scale experiment showed that the CWs-based mechanism can be effective for removing NH4+-N from groundwater. Do you think that this technology can be easily (in terms of cost and energy) and effectively reproducible at the field-scale? Please discuss this into the final discussion session.
In this study, the potential and the effectiveness of the CWs-based mechanism in removing NH4+-N from groundwater are well discussed and presented. I think that the main advantages of this technology include its sustainability and its low impact on the environment. This work deserves publication, but requires some revisions in the Material and Methods overall. I suggest to re-write in a best way this part as a reader has to understand the methodology throughout the test.
Author Response
Thank you very much for reviewing our manuscript, offering valuable comments and for your judgment on our manuscript. We have revised our manuscript following your comments and suggestions. The modified parts are colored in blue. We are happy if you kindly consider that our revision is enough to meet your comments.
Comment 1:
The structure of the Materials and Methods section needs to be improved. I found it difficult to follow the preparation of the experimental material as well as clearly understand the analysis procedure. In order to enhance the understanding of the experimental setup, Figure S1 may be inserted within the manuscript (in Sections 2.2.1 and 2.2.2) and not as supplementary material. Moreover, I think that adding flowcharts in Sections 2.2.1, 2.2.2 and 2.4 could facilitate the reader to fully understand and follow the description of the various steps of the experiment.
Answer 1:
Thank you for your suggestions. We inserted Figure S1 in the manuscript as Figure 1. We have made the flowcharts for Section 2.2.1, 2.2.2, 2.2.3, and 2.4 and added them in the supplementary file (as Figure S1) to clarify and understand the various steps of experiment in this study.
Comment 2:
Lines 149 and 152: “One hundred mg” and “Fifty mL” should be replaced with numbers.
Answer 2:
We revised Line 162: Zeolite (100 mg) and Line 164-165: Synthetic NO3‒-N contaminated groundwater (50 mL).
Comment 3:
Results derived from the laboratory-scale experiment showed that the CWs-based mechanism can be effective for removing NH4+-N from groundwater. Do you think that this technology can be easily (in terms of cost and energy) and effectively reproducible at the field-scale? Please discuss this into the final discussion session.
Answer 3:
Thank you for your suggestion.
Tidal flow CW can be operated by using pump or siphon in field-scale. There are some previous studies on pilot-scale tidal flow CWs. We will carry out the pilot-scale feasibility study. Thus, we explained the feasibility of field-scale tidal flow CW at discussion section (Line 396-399): Pilot-scale tidal flow CWs have been operated by pump- or siphon-driven for treatment of various wastewaters [26, 51, 54, 55]. Like these studies, it will be necessary to evaluate the performance of tidal flow CW treating NH4+-N contaminated groundwater in a pilot- or full-scale system.
Reviewer 2 Report
The authors are suggested to clarify the following issues, while revising their manuscript:
In this study, the ability and characteristics of zeolite-based tidal flow CWs to remove N from NH4+-N contaminated groundwater were examined. However, no chemical analysis or microscopic analysis of the zeolite used (before and after treatment) is included in the study. Unless this is only a performance study and not explicit study on the ability and characteristics of zeolite-based tidal flow CWs to remove N, as it is stated on the title and the scope of the study, then these analysis should be included.
In line 314 the authors state that “Tidal flow CWs have a higher atmospheric air supply and enhanced nitrification and denitrification compared to continuous flow CWs”. The aerobic nitrification and anaerobic denitrification can occur in close proximity in CWs. Atmospheric oxygen diffuses through the floodwater creating an aerobic layer with formation of NO3-. NO3- diffuses to the underlying anaerobic layer, where it can be denitrified. Identification of the oxidized and reduced layers can be done through measurement of a redox potential profile. Such measurement would have distinguished the zones and strengthen the authors assumption.
In line 358, the authors state that “The highly efficient and sustainable N-removal in tidal flow CWs might be due to the regeneration of zeolite NH4+-N adsorption capacity”. NH4+-N adsorption kinetics and sorption isotherms would have validated this statement.
In line 148-157, the authors describe the simulated denitrification experiments, using common reed roots from tidal flow CWs in vials. They state that all vials were purged with nitrogen (N2) gas for 2 minutes to create anaerobic condition. To properly simulate the conditions that occur inside the CWs, changes of DO content in CWs along with depth could have been determined e.g. by using an oxygen microsensor. Such measurements, along with the microbial profile analysis would have accurately identified the different zones present in the CWs and the exact occurrence of denitrification.
The transfer of Fig.S1 (the experimental setup) from supplementary material to the manuscript would also be helpful.
It is a study aiming to examine the ability and characteristics of zeolite-based tidal flow CWs to remove N from NH4+-N from contaminated groundwater.
The study has many strengths and is interesting to reading. Its limitations that need further clarification are included in the authors’ section.
Author Response
Thank you very much for reviewing, careful consideration and judgment on our manuscript. We have revised our manuscript following your comments and suggestions. The modified parts are colored in blue. We are happy if you kindly consider that our revision is enough to meet your comments.
Comment 1:
In this study, the ability and characteristics of zeolite-based tidal flow CWs to remove N from NH4+-N contaminated groundwater were examined. However, no chemical analysis or microscopic analysis of the zeolite used (before and after treatment) is included in the study. Unless this is only a performance study and not explicit study on the ability and characteristics of zeolite-based tidal flow CWs to remove N, as it is stated on the title and the scope of the study, then these analysis should be included.
Answer 1:
Thank you very much for your suggestions. Unfortunately, in this study, we did not analyze chemical and surface characteristics of zeolite. We examined the NH4+-N adsorption kinetic constant of zeolite before and after treatment. We clearly show the adsorption kinetics at Line 245-247, 346-349, Figures 3 and 4, and Figure S2.
According to the comment, we will examine the characteristics of zeolite in more detail in the next experiment.
Comment 2:
In line 314 the authors state that “Tidal flow CWs have a higher atmospheric air supply and enhanced nitrification and denitrification compared to continuous flow CWs”. The aerobic nitrification and anaerobic denitrification can occur in close proximity in CWs. Atmospheric oxygen diffuses through the floodwater creating an aerobic layer with formation of NO3-. NO3-diffuses to the underlying anaerobic layer, where it can be denitrified. Identification of the oxidized and reduced layers can be done through measurement of a redox potential profile. Such measurement would have distinguished the zones and strengthen the authors assumption.
Answer 2:
We understand and agree with your suggestions. Unfortunately, in this study, we did not monitor dissolved oxygen or redox potential in our CWs. However, previous study reveals the oxidized/reduced or aerobic/anaerobic changes in tidal flow CWs. Therefore, we added sufficient citations and explained at Line 354-359: Oxygen can be rapidly replenished into CW beds by the rhythmic tidal flow, and then dissolved oxygen (DO) is consumed by microbial activities for organic carbon degradation and nitrification along the depth [42, 43] and contacting time [44, 45]. After that, CW beds can change into anaerobic conditions. Unfortunately, DO concentration or oxidation-reduction potential was not monitored in this study. Enhanced nitrification and denitrification might have resulted in the in-situ biological regeneration of zeolite in the tidal flow CWs.
Comment 3:
In line 358, the authors state that “The highly efficient and sustainable N-removal in tidal flow CWs might be due to the regeneration of zeolite NH4+-N adsorption capacity”. NH4+-N adsorption kinetics and sorption isotherms would have validated this statement.
Answer 3:
Thank you for your suggestion. In this study, we calculated NH4+-N adsorption kinetic constants of zeolites. We clearly show and explained the kinetic constants at Line 245-247, 346-349, Figures 3 and 4, and Figure S2. However, we did not examine the adsorption isotherms. We appreciate your understanding.
Comment 4:
In line 148-157, the authors describe the simulated denitrification experiments, using common reed roots from tidal flow CWs in vials. They state that all vials were purged with nitrogen (N2) gas for 2 minutes to create anaerobic condition. To properly simulate the conditions that occur inside the CWs, changes of DO content in CWs along with depth could have been determined e.g. by using an oxygen microsensor. Such measurements, along with the microbial profile analysis would have accurately identified the different zones present in the CWs and the exact occurrence of denitrification.
Answer 4:
Thank you for your suggestion. We also consider that change of DO content in CWs along the depth or time will provide the useful findings to understand nitrification and denitrification in CWs. However, we do not have an oxygen microsensor and could not monitor the DO profile in CWs. However, previous study reveals the oxidized/reduced or aerobic/anaerobic changes in tidal flow CWs. Therefore, we added sufficient citations and explained at Line 354-358.
In future experiments, we will monitor the DO profile and aerobic/anaerobic change in CWs for better understanding.
Comment 5:
The transfer of Fig.S1 (the experimental setup) from supplementary material to the manuscript would also be helpful.
Answer 5:
Thank you for your suggestions. We inserted Figure S1 in the manuscript as Figure 1.
Round 2
Reviewer 2 Report
The authors's made revisions that warrant publication in Water journal.
Author Response
Thank you very much for your careful consideration and judgment of our manuscript water-791485.
Based on the valuable comments and suggestions from the editor, we have revised our manuscript. The revised parts are colored in red. We hope our revisions are enough to satisfy the comments and make the manuscript more effective.
Thank you very much for your understanding.